# The Hepatic Antioxidant System Damage Induced with the Cafeteria (CAF) Diet Is Largely Counteracted Using SCD Probiotics during Development of Male Wistar Rats

**DOI:** 10.3390/nu15214557

**Published:** 2023-10-27

**Authors:** Nurdan Aba, Enver Fehim Koçpınar, Taha Ceylani

**Affiliations:** 1Department of Biology, Science Faculty, Muş Alparslan University, 49250 Mus, Turkey; 2Department of Medical Laboratory Techniques, Vocational School of Health Services, Muş Alparslan University, 49250 Mus, Turkey; 3Department of Food Quality Control and Analysis, Muş Alparslan University, 49250 Mus, Turkey

**Keywords:** antioxidant system, cafeteria diet, SCD probiotics, gene expression, enzyme activity

## Abstract

The cafeteria (CAF) diet, reflective of predominant Western dietary behaviors, is implicated in hastening weight gain, subsequently resulting in health complications such as obesity, diabetes, and cancer. To this end, it is vital to notice the deleterious consequences of the CAF regimen prior to the onset of complications, which is fundamental for early intervention in the context of numerous diseases. Probiotic-derived postbiotic metabolites have gained attention for their antioxidative properties, offering a potential countermeasure against oxidative stress. This research sought to discern the protective efficacy of SCD Probiotics against liver glutathione system damage arising from the CAF diet during developmental phases. Male Wistar rats, from weaning on day 21 to day 56, were categorized into four groups: a control on a conventional diet; a group on a standard diet enriched with SCD Probiotics; a mixed-diet group comprising both CAF and standard feed; and a combination diet group supplemented with SCD Probiotics. Through the application of real-time PCR, enzyme activity assessments, and quantitative metabolite analyses, our findings highlight the CAF diet’s adverse influence on the liver’s antioxidant defenses via shifts in gene expression. Yet, the inclusion of SCD Probiotics mostly ameliorated these harmful effects. Remarkably, the positive regulatory influence of SCD Probiotics on the liver’s antioxidant system was consistently observed, independent of the CAF diet’s presence.

## 1. Introduction

The global landscape has witnessed a marked surge in the rates of an overweight status and obesity in recent times. A critical factor driving this alarming trend is unhealthy nutritional habits, which have been linked to cardiovascular diseases and diverse forms of cancer, even manifesting at early ages [1,2]. Characterized by its nutritionally poor diversity, the Western diet not only encourages weight gain but also represents an epitome of unhealthy eating patterns [2]. The cafeteria (CAF) diet serves as a representative model of the Western diet, fostering rapid weight gain by promoting hyperphagia [3]. Beyond dietary factors, genetic predispositions and spontaneous mutations further underscore the multifactorial etiology of weight gain and obesity [4].

Diving deeper into the metabolic implications of the CAF diet, there is pronounced support for refined carbohydrate consumption, which concurrently increases the production of reactive oxygen species (ROS) and induces the expression of pro-inflammatory cytokines [5]. At the foundational level, obesity intricately affects metabolism by introducing perturbations at both the gene and protein tiers [2]. For instance, a CAF diet is implicated in diminishing nitric oxide synthase phosphorylation while simultaneously augmenting plasma coagulation activity [6]. Consistent scientific discourse emphasizes the association of obesity with heightened risks of cardiovascular diseases, renal disorders, and select forms of cancer [1,7]. The rise in obesity is accompanied by spikes in conditions such as type 2 diabetes, hypertension, and nonalcoholic fatty liver disease [8]. The pathological cascades initiating obesity predominantly steer lipid metabolic dysfunction via adipose tissue expansion [9], inflammation, ROS-mediated oxidative stress, and exacerbated insulin production and resistance [1,10,11]. Specifically, in rats, a CAF diet amplifies ROS production within the gastrointestinal tract [11] and truncates gut microbiota diversity [12]. Long-term ROS exposure can cause dysbiosis in the intestinal microbiota, potentially leading to intestinal injuries, colorectal cancer, enteric infections, and inflammatory bowel diseases [11,13].

Transitioning to potential interventions, probiotics have emerged as potent modulators of gastrointestinal health. Their salutary influence is predominantly attributed to postbiotic metabolites, many boasting antioxidative properties, and they serve as formidable ROS regulators [14]. Empirical evidence from both in vivo and animal studies supports the notion that food-derived antioxidants have protective effects against oxidative stress in the gut microbiota [13]. While the nexus between diet and the intestinal tract has been thoroughly explored, delving into their broader metabolic impacts can illuminate early-stage metabolic shifts and predict long-term disease onset. Such insights have profound implications for proactive disease prevention and timely obesity-related therapeutic interventions.

In light of the burgeoning global health concerns related to Western dietary patterns, particularly the CAF diet, understanding its molecular impact on vital organ systems has become paramount. Given the emergent antioxidative properties of postbiotic metabolites from probiotics, this study aimed to unravel the depth of damage that the CAF diet inflicts upon the hepatic glutathione system during pivotal development stages. Our guiding hypothesis was that SCD Probiotics, owing to their antioxidative properties, could potentially offset the oxidative disruptions triggered with the CAF diet. Through rigorous molecular and metabolic evaluations, we aimed to validate this hypothesis and highlight the broader implications of dietary interventions for mitigating the risks associated with prevalent Western dietary habits.

## 2. Material and Methods

### 2.1. Experimental Design and Animal Care

Twenty-one-day-old male Wistar rats during the development period were used as the model organism in this study. Weaned rats were randomly divided into four groups. The control group was fed only a rodent diet (n = 7), the CAF group was fed a cafeteria (CAF) diet and a normal rodent diet (n = 7); the combined (Prob + CAF) group was fed a combined diet consisting of CAF and SCD Probiotics (n = 7), and the Prob group was fed a standard rodent diet and SCD Probiotics supplementation (n = 7). As in a previous study, the tests and analyses in the present study were performed with three replicates of samples prepared from at least three tissues selected randomly. To ensure homogeneity, each tissue was completely chopped into tiny pieces by mincing with a sterile scalpel. The samples were prepared using the required amounts of these homogenous solid tissues [15,16]. SCD Probiotics are a product of a food manufacturing company and are used as a food supplement (Essential Probiotics XI—500 mL H.S. Code: 2206.00.7000). The treatments were continued until day 56 at the end of the development period. The SCD probiotic supplement was administered with oral gavage at a dose of 3/2 mL (1 × 10^8^ CFU) per day [17]. SCD Probiotics include *Bacillus subtilis*, *Bifidobacterium bifidum*, *Bifidobacterium longum*, *Lactobacillus acidophilus*, *Lactobacillus bulgaricus*, *Lactobacillus casei*, *Lactobacillus fermentum*, *Lactobacillus plantarum*, *Lactobacillus lactis*, *Saccharomyces cerevisiae*, and *Streptococcus thermophilus*. Animals were fed ad libitum with a standard rodent diet, and a cafeteria diet was provided in addition to the normal ad libitum feeding. Throughout the experiment, the animals’ weight, weekly food consumption, cafeteria diet content, and total energy values were recorded (Table 1). On the 56th day of administration, the animals were sacrificed immediately after being slightly dazed using ether treatment. Liver tissues were extracted, shocked on dry ice, and stored in a −80 °C deep freezer until the analysis. All animals were housed in accordance with the standard animal care protocols. This study was approved by the Ethics Committee (meeting date: 29 June 2021, approval number: 2021/03) of the Bingöl University Animal Experiments Local Ethics Committee.

### 2.2. Determination of the Quantitative GSH Level and GSH/GSSG Ratio

The quantitative GSH level was determined according to the method of Griffith (1980) [18], modified slightly by Sonmez Aydin, et al. [19]. The principle of this method is based on the oxidation of GSH with sulfhydryl reagent DTNB at a wavelength of 412 nm. In total, 100 mg of tissue was weighed, and the homogenization process was performed in 500 µL of a trichloroacetic acid (TCA) solution (5% *m*/*v*) using a TissueLyser LT (Qiagen, Hilden, Germany) homogenizer. The homogenates were centrifuged at 13,000 rpm for 10 min at 4 °C and divided into two parts to measure total quantitative GSH and GSSG levels. GSSG measurement was performed after derivatization of the current GSH using 2-vinylpyridine (QP2VP) for 1 h at room temperature. Quantitative GSH and GSSG levels were recorded by measuring kinetically at 412 nm [19]. Standard samples were prepared using pure GSSG at 10 different concentrations (1 µg/mL–10 µg/mL) and the absorbance values were measured at 412 nm. A standard curve graph was created using 10 different GSSG solutions. Total quantitative GSH and GSSG levels were calculated using the GSSG standard curve and GSH/GSSG was calculated by proportioning the results obtained.

### 2.3. Determination of Lipid Peroxidation

Malondialdehyde (MDA) was determined according to the method of Ohkawa, et al. [20]. Approximately 50 mg of tissue was weighed, and homogenization was performed in 500 µL of a 50 mM Tris-HCl buffer (pH: 7.2) for 10 min. To increase the effective adhesion to thiobarbituric acid (TBA), the proteins were precipitated using 10 mL of a KCl solution (1.15% *m*/*v*) for each 0.1 absorbance value observed in Bradford results. Homogenates were centrifuged at 16,000 rpm for 3 min and precipitation was discarded. Absorbance values were read using a microplate reader (Multiscan GO, Thermo Scientific, Waltham, MA, USA) at 532 nm.

### 2.4. RNA Isolation and cDNA Synthesis

RNA isolation was carried out following the instructions of the ThermoFisher-labeled PureLink RNA Mini Kit protocol (Catalog Number: 12183018A). The purity and concentration of the product were checked using the NanoDrop QC Skanlt software 4.1 feature of the 96-well plate spectrophotometer (Multiskan GO, Thermo Scientific, Waltham, MA, USA). cDNA synthesis was carried out following the instructions of the ProtoScript First Strand cDNA Synthesis Kit (NEW ENGLAND BioLabs, E6300L, Ipswich, MA, USA). The thermal cycle was performed as follows: denaturation (5 min at 70 °C), incubation (1 h at 42 °C), and enzyme inactivation (5 min at 80 °C) using a Sensoquest Thermocycler Labcycler w/Thermoblock 96 Gold Plated Silver 012-103, Germany by Hannah-Vogt-Str.1 (cDNA was stored at −20 °C).

### 2.5. Primers and Gene Expression Analysis

The primers were designed with the help of the National Center for Biotechnology Information (NCBI) database. The *gapdh* gene was used as a reference gene in the qPCR application. The alterations in antioxidant system genes were determined with the SYBR Green Qpcr method (quantitative real-time polymerase chain reaction). qPCR reactions were carried out on the Rotor-Gene Q instrument (QIAGEN, Inc., Hilden, Germany) in strip tubes (0.1 mL, 4-well). The final volume of the reaction mixture for a single strip tube was prepared as 10 μL. One cycle (50 °C for 2 min, 95 °C for 1 min) and forty cycles (95 °C for 10 s, 60 °C for 30 s) were used as a thermal profile. qPCR CT values were converted to the relative mRNA expressions using the 2^−ΔΔCT^ calculation method proposed by Livak and Schmittgen [21]. NCBI accession numbers and primer sequences are shown in Table 2.

### 2.6. Homogenate Preparation and Protein Determination Assay

In total, 100 mg of tissue was weighed for all enzyme activity measurements and homogenized in 1 mL of the buffer used specific to each enzyme. Homogenization was performed using a TissueLyser LT (Qiagen) homogenizer at 50 Hz for 3 + 3 min as 2 periods. Homogenates were centrifuged as expressed in enzyme activity assays and precipitations were discarded. Quantitative amounts of proteins in supernatants were determined with Bradford’s protein assay [22]. By diluting 1 mg/mL stock bovine serum albumin (BSA), standard samples were prepared at 10 different concentrations (20 µg/mL–200 µg/mL) and a standard curve was prepared to calculate quantitative protein amounts. Absorbance values were recorded at 595 nm. Quantitative protein amounts were determined using the standard curve equation.

### 2.7. Enzyme Activity Assays

Homogenization was performed in a 50 mM potassium phosphate buffer (pH: 7.4) using a TissueLyser LT (Qiagen) homogenizer at 50 Hz for 3 + 3 min as 2 periods. Homogenates were centrifuged at 10,000 rpm for 1 h and transparent homogenates were used to determine the activities of SOD and CAT enzymes. CAT enzyme activity was measured at 240 nm and 15 s intervals for 1 min [23]. SOD enzyme activity was measured at 560 nm as a fixed wavelength [24]. Tissues were homogenized again to measure the activities of GR, GPx, and GST in 50 mM Tris-HCl (pH: 7.6) including 1 mM DDT (dithiothreitol), 1 mM EDTA (ethylenediaminetetraacetic acid), and 1 mM PMSF (phenylmethanesulfonylfluoride). The lysate was centrifuged at 13,000× *g* for 1 h and supernatants were used for spectrophotometric measurements of GR, GPx, and GST activities. GR activity was measured using the method of Carlberg and Mannervik [25] for 3 min and 340 nm. GPx activity was measured according to the Sigma Aldrich experimental protocol (Cat. No.: SPGLUT02). According to the protocol, the reaction mixture was first incubated for 10 min to stabilize the absorbance changes. Then, H_2_O_2_ was added and GPx activity was measured kinetically for 5 min and 340 nm. GST activity was measured for 3 min and 340 nm using the method of Habig, et al. [26]. Enzyme activities were first calculated as EU/mL and then specific activities, by proportioning to Bradford’s results, were calculated as EU/mg of protein (Equation (1)). The components of Equation (1) are expressed as A: Absorbance changes per min, Ɛ: The extinction value of the substrate, V_T_: Total volume, V_S_: Sample volume, DF: Dilution factor.
(1)$$\left(EU/mg\right)=\frac{A}{Ɛ\cdot t}\times \frac{V_{T}}{V_{S}}\times DF\times \frac{1}{mg\;protein/mL}$$

### 2.8. Data Collection from Databases and the Validation of Obtained Data

The Gene Expression Profiling Interactive Analysis (GEPIA) is an interactive web server that provides normal and tumor sample data via Genotype-Tissue Expression (https://www.gtexportal.org/) and the University of Alabama at Birmingham Cancer data analysis Portal (UALCAN) (https://ualcan.path.uab.edu/) databases. Excessive weight is linked to numerous diseases, and the CAF diet is a robust and reliable diet model inducing weight gain and obesity. The data show the relationship between the human antioxidant system and obesity-mediated hepatocellular carcinoma using the GEPIA [27] and UALCAN [28] databases. First, the gene expression profile of the hepatic antioxidant system in human liver hepatocellular carcinoma (LIHC) was accessed via the link http://gepia.cancer-pku.cn/ and the regulation of proteins in human hepatocellular carcinoma (HCC) was accessed via the link https://ualcan.path.uab.edu/ (accessed on 5 July 2023). Based on database information, we interpreted the gene and protein expressions of the hepatic antioxidant system.

### 2.9. Statistical Analysis

All measurements were repeated at least 3 times. Subsequently, the results from diet groups were statistically compared to the results of the control groups. All statistical data were shown as the Mean ± SEM (standard error mean). The results were statistically compared using Tukey’s Multiple Comparisons Test following One-way ANOVA, and *p* < 0.05 was considered significant. The relationship between mRNA expressions and enzyme activities based on average body weight change (AWC) was evaluated using the nonparametric Sparman’s correlation analysis. The compatibility and direction of correlation were expressed by calculating the *p* and r values, respectively. Statistically significant differences were represented in the following way: ^ns^
*p* > 0.05 (not significant); * *p* < 0.05 (significant); ** *p* < 0.01 (very significant); *** *p* < 0.001 and **** *p* < 0.0001 (extremely significant).

## 3. Results

### 3.1. Effects of Different Diet Combinations on Quantitative Hepatic Metabolite Levels

The measurement of MDA serves as a critical marker showing the extent of lipid peroxidation in a cell. Consequently, the determination of MDA holds significance in the acquisition of data pertaining to a series of issues, including oxidative stress, toxicity, and cancerization [16]. Various dietary regimens were administered to male Wistar rats during their development period for 5 weeks by focusing on assessing their primary impact on the hepatic antioxidant system prior to the onset of obesity. In this context, quantitative MDA concentrations and *Foxo1* mRNA expression levels were investigated. Consequently, the consumption of the CAF diet led to a significant elevation in MDA concentrations. However, a remarkable reduction to a lower level than the control group was observed when SCD Probiotics were administered. In addition, the MDA concentrations in the Prob group were detected to be relatively lower than the control group. A relative decrease in the expression level of the *Foxo1* gene was observed in the CAF group and the supplementation of SCD Probiotics did not regulate the effect of the CAF diet (Figure 1A).

Reduced glutathione (GSH) and oxidized glutathione (GSSG), which are determinants of cellular redox status, are important metabolites to maintain the reduced environment in a cell. Based on the importance of the reduced environment, GSH generation is the cell’s primary choice to maintain the cellular glutathione pool. Hence, the cellular glutathione system plays a pivotal role in controlling the levels of both reduced glutathione (GSH) and oxidized glutathione (GSSG) and due to the link of the deviations in the GSH concentration and GSH/GSSG ratio with oxidative stress and cancer, firm regulation of the GSH/GSSG ratio is essential for maintaining the balance within the antioxidant system [16]. For this reason, the effects of various diet combinations on the concentration of GSH and GSSG were investigated. Strikingly, the levels of the GSH, GSSG, and GSH/GSSG ratio increased significantly in the CAF group (Figure 2A, B, and C, respectively). The GSH level in the combined diet group decreased with the addition of SCD Probiotics supplementation, whereas GSH levels increased when SCD Probiotics were administered alone (Figure 2A). Ultimately, all metabolites increased in the CAF group, but SCD Probiotics supplementation fully reversed this effect. Notably, SCD Probiotics had a more pronounced impact in the combined diet group compared to its administration alone.

### 3.2. Effect of Diet Practices on Hepatic Antioxidant Gene Expressions

The alterations induced with dietary practices in gene expressions within the hepatic antioxidant system were quantified using quantitative real-time PCR (qPCR). Although there was an increase in *Gst* gene expression, statistically significant decreases were observed in the expression of other genes (Figure 3). The gene expression levels of *Cat*, *Gpx*, and *Gr* were relatively reversed with SCD Probiotics in the combined diet group (Figure 3B, C, and D, respectively), but the regulation in expression levels was found to be insufficient compared to the control group. Also, statistically significant change was not observed when SCD Probiotics were applied alone, except for the increase in *Cat* expression (Figure 3).

### 3.3. The Effect of Dietary Practices on the Enzyme Activities of Hepatic Antioxidant System

Enzyme activities of the hepatic antioxidant system members localized in the cytoplasm including SOD, CAT, GPx, GR, and GST were spectrophotometrically measured in control and experimental groups. All enzyme activity results were found to be largely consistent with gene expression results, except for GR activity (Figure 4). Contrary to the significant increases in GR and GST activities (Figure 4D,E), the CAF groups exhibited significant decreases in SOD, CAT, and GPx activities (Figure 4A–C). Although SCD Probiotics relatively regulated the activities of all proteins in the combined diet groups, except for GST activity, the regulation seems to be insufficient when compared to the controls. Strikingly, a dramatic decrease to a lower level than the control group was observed in the GST enzyme activity (Figure 4E). When SCD Probiotics were applied alone, the activities of CAT and GPx increased significantly, but there was no significant change in SOD, GR, and GST activities (Figure 4).

### 3.4. Correlation of Hepatic Antioxidant System Based on AWC

The comparative interpretation of gene expressions and enzyme activities is required to better understand the indirect relationship between weight gain and HCC. To achieve this, the changes of up or down changes in the gene expressions and enzyme activities based on weight gain were analyzed with nonparametric Spearman’s correlation test. The strength and direction of correlation were displayed by calculating *p* and r values. r values close to −1 and +1 indicate the presence of negative and positive correlation, respectively. Contrary to the increase in *Gst* expression, inverse correlation based on AWC was observed in other gene expressions and less significant correlations were observed in the combined diet group. Administration of SCD Probiotics alone resulted in relative weight reduction and mostly increased enzyme activities. This means that there is an inverse correlation between enzyme activity and AWC (Figure 5).

### 3.5. Identification of LIHC Prevalence Based on Obesity

The CAF diet is an unhealthy diet that includes fried food, processed food, and fast food. For this reason, it promotes weight gain more rapidly than traditional diets and is widely used to create the animal models of obesity [2,7]. It is well known that there is a relationship between obesity and cancer. As a supporter of this, the expression profiles of the antioxidant system were defined in healthy and liver hepatocellular carcinoma (LIHC) using the GEPIA and the UALCAN databases [27,28]. Database information reveals a decrease in *Foxo1* and *Cat* gene expressions but increases in other gene expressions in LIHC. GEPIA and UALCAN databases showed a decrease in the protein expressions of all hepatic antioxidant systems. The results of the current study were evaluated based on the data provided from GEPIA and UALCAN databases (Figure 6).

### 3.6. Identification of LIHC Prevalence Based on Patient Weight

The UALCAN database was used to identify the changes in hepatic antioxidant gene expressions based on the relationship between patient’s weight and LIHC [28]. The UALCAN database reveals the general increase in *Gpx* and *Gr* expressions and general decrease in the expressions of the other genes in LIHC (Figure 7). The present study examined the impact of dietary practices on the hepatic antioxidant system in male Wistar rats during their developmental period. To interpret the effects of dietary practices more clearly, the results of the present study were compared with the information obtained from the database. Consequently, a significant similarity in gene expression changes was observed, except for *Gpx* and *Gst* expressions.

## 4. Discussion

The effects of dietary practices, including a cafeteria (CAF) diet and SCD Probiotics, on the hepatic antioxidant system were investigated in male rats during their developmental period. In the present study, metabolite levels, gene expression, and enzyme activities were determined. The CAF diet led to statistically significant increases in various metabolite levels (MDA, GSH, and GSSG), and SCD Probiotics effectively reversed this effect. Unusual changes in MDA and GSH levels are crucial indicators of lipid peroxidation and concretization [16,29]. While it mostly had no significant effect when administered alone, SCD Probiotics notably reversed the impact of the CAF diet. Based on these data, it can be said that SCD Probiotics exhibited an aggressive response to the CAF diet in the combined diet group. In addition, SCD Probiotics supplementation alone reduced MDA concentrations (Figure 1B). This is a critical result, emphasizing its role as a lipid peroxidase suppressant and regulator. 

Molecules such as H_2_O_2_ and free radicals act as oxidation mediators, and oxidative stress is a critical consequence of excessive ROS concentration at uncontrollable levels [15]. The oxidation of molecules with excess ROS can trigger chain oxidation reactions by affecting new molecules. Consequently, structural degeneration within cells is considered a precursor to current diseases such as cancer, Parkinson’s disease, and Alzheimer’s disease, which are closely linked to cellular communication. A certain amount of cellular ROS plays a vital role in the healthy progression of this communication, and the antioxidant system serves as crucial insurance to maintain the ROS balance in the thin line [16]. The required ROS are produced from O_2_ entering the cell, without any manipulation, and signal transduction mechanisms progress with ROS regulated by the antioxidant system, including SOD, CAT, GPx GR, and GST, and metabolites such as GSH and GSSG. The primary choice of a healthy cell to maintain redox balance by maintaining a reduced environment is to support the GSH pool [30,31]. Changes in cellular GSH and the GSH/GSSG ratio are associated with oxidative stress and cancer development [16]. Previous studies have also reported that elevated GSH levels enhance resistance to oxidative stress by boosting antioxidant capacity, which has been observed in some types of cancer [29]. The observed elevations in GSH, GSSG, and GSH/GSSG may be the response of a background defense mechanism (Figure 2A–C). That is, the reason for the elevation in the GSH level may be a response to prevent the possibility of HCC associated with weight gain induced with CAF. The elevated MDA in the CAF group supports this hypothesis. SCD Probiotics reversed the effect of CAF, and the GSH pool seems to be supported when SCD Probiotics were applied alone. The increase in GSH levels may also be a response that reduces MDA concentration induced with CAF. The CAF diet is a robust and reliable method inducing obesity in animals [4] and obesity is known to carry HCC risk [32,33]. Ceylan [32] emphasized that the expression levels of 33 genes change in both obesity and HCC, and a series of genes, including *Foxo*, are interrelated. *Foxo1* contributes to apoptosis by inducing its markers and apoptosis is a blocker of the cancerization process [32]. *Foxo1* expression has also been reported to be decreased in HCC [34]. Hence, we investigated the impact of dietary practices on *Foxo1* expression in the present study and found that the CAF diet reduces *Foxo1* expression (Figure 1A).

Upward or downward changes in SOD and CAT expression have been reported depending on the cancer type [35,36]. For this reason, the effects of dietary practices on hepatic antioxidant system genes were investigated, and the experimental findings indicated a statistically significant decrease in the expression levels of Sod and Cat in the group exposed to CAF (Figure 3A and B, respectively). SCD probiotic supplementation had no significant effect on *Foxo1*, but it was able to partially reverse the expressions of *Sod* and *Cat* (Figure 3A and B, respectively). Based on the present study, the CAF diet seems to contribute to the possibility of carcinogenesis in liver tissue (Figure 1A). In the CAF group, we observed significant reductions in Gpx and Gr expression, and a relative increase in *Gst* expression (Figure 3C, D, and E, respectively). When SCD Probiotics were added, the impact of the CAF diet on Gr expression was fully eliminated, and the expression of *Gpx* and *Gst* was partially restored. When SCD Probiotics were administered alone, gene expression of the hepatic antioxidant system was not affected. Previous studies have highlighted decreases in the expression of antioxidant proteins because of oxidative stress [37,38]. The overall reduction in the gene expression of the hepatic antioxidant system indicates the possibility of cancerization induced with CAF. In this context, our current study also indicates that SCD Probiotics may be an important blocker of oxidative stress, which emphasizes the crucial role of daily probiotic consumption in youth.

The present study also aimed to examine the impact of the administered dietary regimens on the enzyme activities of the hepatic antioxidant system. In the CAF groups, contrary to the significant increases in GR and GST activities, statistically significant reductions were observed in the activities of other enzymes. Following SCD probiotic supplementation, the activities of all antioxidants were regulated at different rates and the most notable regulation was observed in GST activity. When SCD Probiotics were applied alone, SOD, GR, and GST activities were not affected, but CAT and GPx activities were increased (Figure 4). Despite the aggressive response of SCD Probiotics to the CAF diet, its benign behavior is remarkable when applied alone. This may be due to the regulatory effects of postbiotic metabolites on SCD Probiotics. This observation in the kinetic enzyme activity study suggests that the CAF diet may act as a peroxidation inducer, and SCD Probiotics could serve as a peroxidation scavenger.

To better understand the effects of dietary practices on the hepatic antioxidant system, by using the GEPIA and UALCAN databases, the changes in gene and protein expressions of the human hepatic antioxidant system based on HCC were identified, respectively (Figure 6). The GEPIA database showed the increases in the expression of *Foxo1* and *Cat*, and the decreases in the expression of other genes. The changes in the expression of *Cat*, *Gst*, and *Foxo1* were in line with the findings from the GEPIA database. Furthermore, the UALCAN database analysis clearly identifies a discernible reduction in hepatic antioxidant system protein expression in HCC, and the enzyme activities of SOD, CAT, and GPx are in line with UALCAN database definitions. Because of the link between the CAF diet and weight gain, we, using the UALCAN database, identified the changes in human hepatic antioxidant gene expression in LIHC based on the patient’s weight [28]. It is clear from the UALCAN database that there are noticeable increases in Gpx and Gr expression, and decreases in other genes in patients with LIHC based on body weight (Figure 7). Interestingly, a decrease in Gpx and Gr expression was detected in the present study. This could be considered an early genetic response or a new regulation occurring prior to the onset of a complication like obesity in the development period of male Wistar rats.

To monitor the gradual regulation of the hepatic antioxidant system, a correlation analysis based on AWC was performed (Figure 5). Remarkable compatibility was observed in gene expression and enzyme activities of the hepatic antioxidant system in the CAF group. It is also pertinent to highlight that despite the absence of a statistically significant alteration in AWC, notable alterations in mRNA expression and enzyme activities were observed. This confirms CAF’s harmful potential. Also, the examination revealed that the alterations in mRNA expression and enzyme activities were not exactly in the same line. This may be due to epigenetic modifications released as a response in the cell. Natural regulation that occurs genetically is essential for the continuation of a cell’s optimal life. However, endogenous, or exogenous, changes may trigger epigenetic responses by affecting the cellular metabolic reactions and ultimately it is possible for epigenetic responses to occur at different levels. In this case, the reduction in SOD and CAT activities may be early epigenetic indicators against cancerization (Figure 4A and B, respectively), and GPX and GR activities that support GSH production may be a response to the possibility of oxidative stress that can induce HCC (Figure 4C and D, respectively). The increase in GST activity may also be a precursor to a possible toxicity induced with CAF. In brief, changes in GR, GPX, and GST activities may be a novel regulatory response that reduces the effect of the CAF diet.

The gut–liver axis plays a pivotal role in mediating effects on the ROS system. The cardinal function of an intact intestinal barrier restricts the translocation of potentially noxious substances into systemic circulation. Nevertheless, compromised intestinal integrity paves the way for bacterial endotoxins, specifically lipopolysaccharides (LPSs), to permeate the bloodstream. When these endotoxins reach the hepatic tissues, they serve as potent catalysts for inflammation, thereby amplifying oxidative stress [39]. Interestingly, recent scientific endeavors have shed light on the potential of probiotic and prebiotic supplements to confer hepatoprotective effects. This protection is postulated to arise from modulation of the gut microbiota [40]. In a seminal investigation involving 24-month-old male Sprague–Dawley rats, the combination of SCD Probiotics and TUDCA was examined. These findings are consistent. There was a pronounced enhancement in the alpha diversity indices of the gut microbiota following the 30-day regimen of SCD Probiotics. In addition, a significant decrease was observed in the Firmicutes to Bacteroidetes (F/B) ratio, a salient biomarker indicative of dysbiosis and an array of metabolic diseases [17].

Specific probiotic strains such as Lactococcus lactis and Bifidobacterium bifidum, which are constituents of SCD Probiotics, have demonstrated pronounced hepatoprotective effects. In a mouse model induced with tBHP, the administration of these probiotic strains led to a notable reduction in hepatic aspartate transaminase, alanine transaminase, and lipid peroxidation levels, underscoring their antioxidant potential [41]. Cumulatively, these findings resonate with the notion that such probiotics, both in in vitro and in vivo models, manifest substantial antioxidant activities, paving the path for therapeutic applications in hepatic health. The fact that a similar or even stronger effect than the use of probiotics is achieved with a fasting diet shows that metabolism requires a fasting state as well as satiety. It was found that the dysbiosis in the intestinal microbiota of 12-month-old male Wistar rats was resolved and individual alpha diversity indices increased after a 30-day intermittent fasting program [42]. Similarly, 1-month intermittent fasting programs have been found to provide a rejuvenating effect on the biomolecular structure of liver and heart tissue and significantly reduce protein carbonylation, a marker of oxidative stress [43,44].

## 5. Conclusions

In conclusion, the CAF diet led to a decrease in the expression of almost all hepatic antioxidant system components, except for both expression levels of GST and GR activity. The expression of *Foxo1*, *Sod*, and *Cat* is downregulated in HCC. Using the UALCAN database, the expression of *Foxo1*, *Sod*, *Cat*, and *Gst* was also found to decrease based on the patient’s weight and LIHC. According to the present study, *Foxo1*, *Sod*, and *Cat* may be among the first genes affected before the onset of HCC induced with the CAF diet, and the CAF diet seems to be one of the agents inducing these changes. However, regulation supporting the production of GSH may be a resistance to HCC formation in the hepatic antioxidant system during the development period of Wistar rats. The results observed for the kinetic enzyme activities support this idea more clearly. Regardless of metabolic regulation, SCD probiotic supplementation partially reduced the adverse effects of CAT; however, full regulation did not occur. The regulatory role of SCD Probiotics in response to the CAF diet and partially beneficial effects on gene expression and enzyme activities when administered alone are noteworthy.

## Figures and Tables

**Figure 1 nutrients-15-04557-f001:**
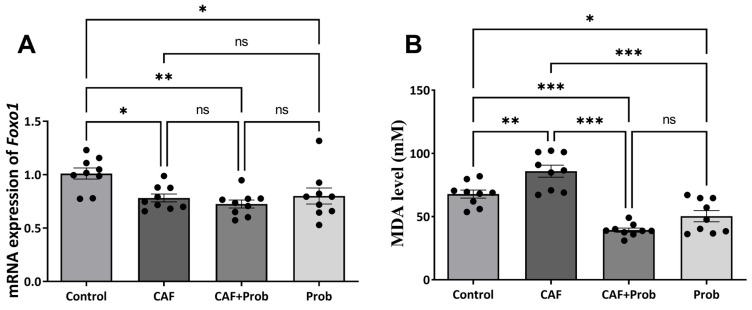
Hepatic MDA level and *Foxo1* mRNA expression in control and experimental groups. (**A**) Relative mRNA expression level of *Foxo1*. (**B**) MDA levels in control and experimental groups. All data are shown as Mean ± SEM (n = 3 and 3 replicates); *p* values are derived using Tukey’s Multiple Comparisons Test followed by One-way ANOVA. Statistically significant differences are indicated in the following way: ^ns^
*p* > 0.05 (not significant); * *p* < 0.05 (significant); ** *p* < 0.01 (very significant); *** *p* < 0.001 and more significant values (extremely significant).

**Figure 2 nutrients-15-04557-f002:**
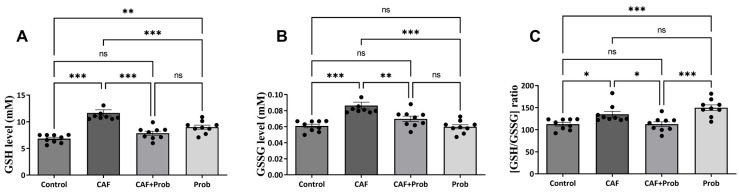
Hepatic metabolite levels and statistical evaluation results. (**A**) Reduced glutathione (GSH) level. (**B**) Oxidized glutathione (GSSG) level. (**C**) Quantitative ratio of GSH to GSSG (GSH/GSSG). All data are shown as Mean ± SEM (n = 3 and 3 replicates); *p* values are derived using Tukey’s Multiple Comparisons Test followed by One-way ANOVA. Statistically significant differences are indicated in the following way: ^ns^
*p* > 0.05 (not significant); * *p* < 0.05 (significant); ** *p* < 0.01 (very significant); *** *p* < 0.001 and more significant values (extremely significant).

**Figure 3 nutrients-15-04557-f003:**
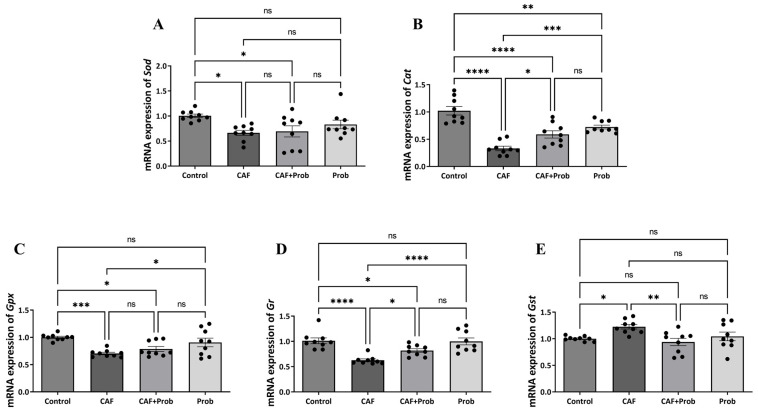
The relative mRNA expression levels in hepatic antioxidant system genes. mRNA, messenger RNA; *Sod*, superoxide dismutase; *Cat*, catalase; *Gpx*, glutathione peroxidase; *Gr*, glutathione reductase; and *Gst*, glutathione S-transferase. (**A**) mRNA expression levels of *Sod*, (**B**) mRNA expression levels of *Cat*, (**C**) mRNA expression levels of *Gpx*, (**D**) mRNA expression levels of *Gr*, and (**E**) mRNA expression levels of *Gst*. All data are shown as Mean ± SEM (n = 3 and 3 replicates); *p* values are derived using Tukey’s Multiple Comparisons Test followed by One-way ANOVA. Statistically significant differences are indicated in the following way: ^ns^
*p* > 0.05 (not significant); * *p* < 0.05 (significant); ** *p* < 0.01 (very significant); *** *p* < 0.001 and **** *p* < 0.0001 (extremely significant).

**Figure 4 nutrients-15-04557-f004:**
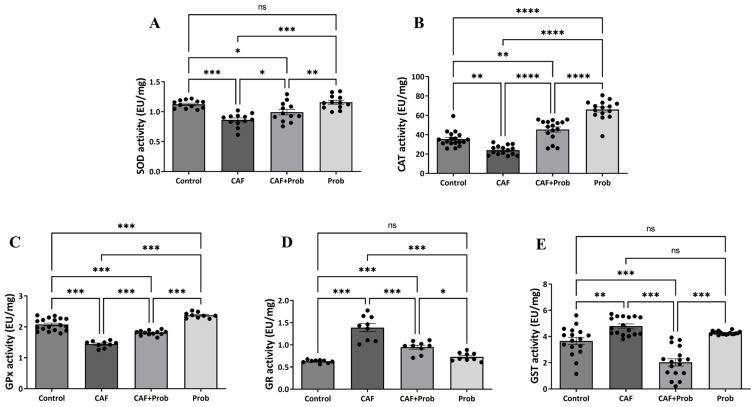
The kinetic enzyme activity results in hepatic antioxidant system. (**A**) Specific enzyme activity of SOD, (**B**) Specific enzyme activity of CAT, (**C**) Specific enzyme activity of GPx, (**D**) Specific enzyme activity of GR, and (**E**) Specific enzyme activity of GST. SOD, superoxide dismutase; CAT, catalase; GPx, glutathione peroxidase; GR, glutathione reductase; and GST, glutathione S-transferase. All data are shown as Mean ± SEM (n = 3 and 3 replicates); *p* values are derived using Tukey’s Multiple Comparisons Test followed by One-way ANOVA. Statistically significant differences are indicated in the following way: ^ns^
*p* > 0.05 (not significant); * *p* < 0.05 (significant); ** *p* < 0.01 (very significant); *** *p* < 0.001 and **** *p* < 0.0001 (extremely significant).

**Figure 5 nutrients-15-04557-f005:**
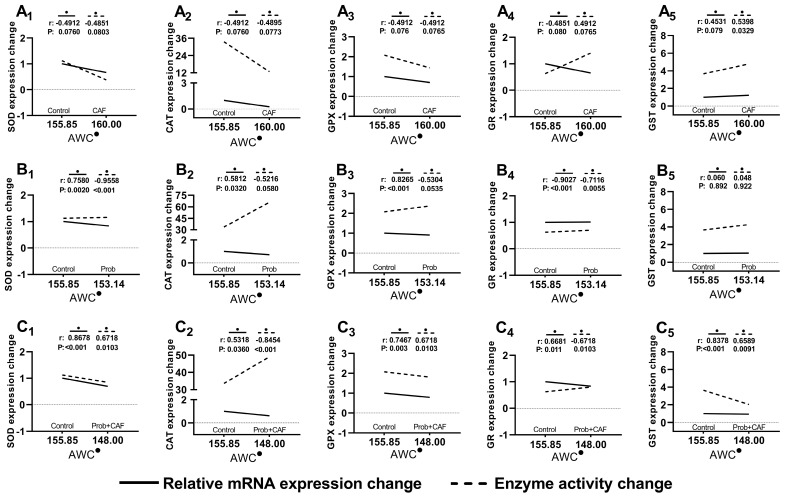
Correlative assessment of mRNA expression and enzyme activities depending on average weight change (AWC) in hepatic antioxidant system. The X-axis shows AWC over 5 weeks. The Y-axis shows the changes in relative mRNA expressions and enzyme activities (EU/mg). (•) is the symbol of AWC in statistical evaluation. The order of group (**A1**–**A5**) shows the expression changes in CAF group, the order of group (**B1**–**B5**) shows the expression changes in Prob group, and the order of group (**C1**–**C5**) shows the expression changes in combined diet (CAF + Prob). The changes in relative mRNA expressions and enzyme activities were evaluated depending on AWC using nonparametric Spearman’s correlation test. r and *p* values are shown on the correlation graphs.

**Figure 6 nutrients-15-04557-f006:**
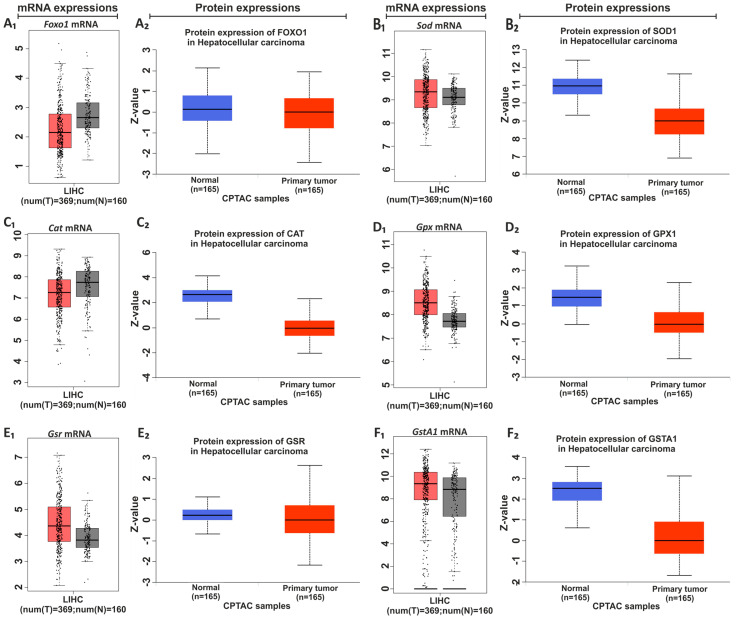
GEPIA and UALCAN data showing mRNA and protein expression changes in HCC. The mRNA and protein expressions of *Foxo1* (**A_1_** and **A_2_**, respectively), *Sod* (**B_1_** and **B_2_**, respectively), *Cat* (**C_1_** and **C_2_**, respectively), *Gpx* (**D_1_** and **D_2_**, respectively), *Gsr* (**E_1_** and **E_2_**, respectively), and *GstA1* (**F_1_** and **F_2_**, respectively). The gray bars in boxplots show normal samples and the red bars in boxplots show tumor samples.

**Figure 7 nutrients-15-04557-f007:**
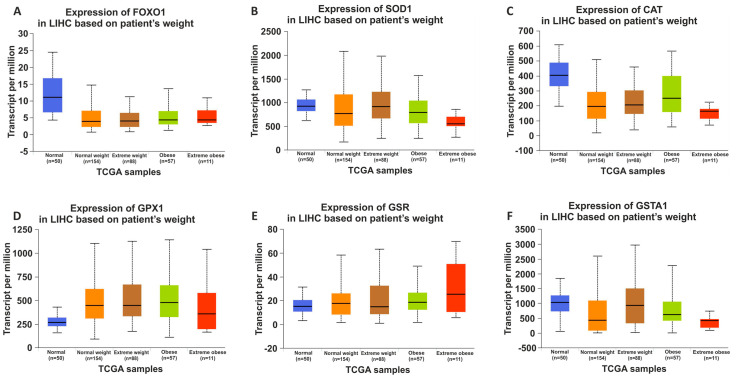
UALCAN data showing mRNA expressions depending on patient’s weight in liver hepatocellular carcinoma (LIHC). mRNA expression changes of *Foxo1* (**A**), *Sod* (**B**), *Cat* (**C**), *Gpx* (**D**), *Gr* (**E**), and *GstA1* (**F**). The blue bars in boxplots show normal samples and other colored bars in boxplots show tumor samples.

**Table 1 nutrients-15-04557-t001:** The amount of food applied, energy values, and average weight changes (AWC) in 5 weeks.

		Onset	Week 1	Week 2	Week 3	Week 4	Week 5
Energy (kcal)	Control	1792	1803	2483	3247	3686	3476
	CAF	1792	2354	3740	5848	6471	6312
	Prob + CAF	1792	3264	3669	5863	5110	6140
	Prob	1792	2093	3056	3629	3575	3285
Rodent nutrition (g)	Control	67	67	93	121	138	130
	CAF	67	61	89	136	134	123
	Prob + CAF	67	38	55	57	50	65
	Prob	67	41	53	58	39	47
AWC	Control	79.9 ± 5.24	105.8 ± 5.61	134.8 ± 7.91	165.6 ± 7.87	196.7 ± 7.26	235.7 ± 6.98
	CAF	79.3 ± 5.24	105.1 ± 5.43	128.6 ± 5.76	173.7 ± 5.96	207.8 ± 7.84	239.3 ± 9.36
	Prob + CAF	79.3 ± 5.69	106.1 ± 5.65	133.1 ± 6.69	172.7 ± 6.05	203.4 ± 7.06	227.3 ± 6.24
	Prob	79.7 ± 5.39	108.1 ± 5.26	130.6 ± 5.87	165.6 ± 7.51	203.7 ± 10.09	232.8 ± 8.78

**Table 2 nutrients-15-04557-t002:** Sequences and accession numbers of specific primers.

Gene Name	Accession Number	Elongation Position	Sequence (5′-3′)
*Superoxide dismutase* (*Sod*)	NM_017050.1	Forward	GCTTCTGTCGTCTCCTTGCT
		Reverse	CTCGAAGTGAATGACGCCCT
*Catalase* (*Cat*)	NM_012520.2	Forward	GCGAATGGAGAGGCAGTGTA
		Reverse	GTGCAAGTCTTCCTGCCTCT
*Glutathione reductase* (*Gr*)	NM_053906.2	Forward	AGTTCACTGCTCCACACATCC
		Reverse	TCCAGCTGAAAGAACCCATC
*Glutathione peroxidase* (*Gpx*)	NM_183403.2	Forward	TGGCTTACATCGCCAAGTC
		Reverse	CCGGGTAGTTGTTCCTCAGA
*Glutathione S-transferase* (*Gst*)	NM_017013.4	Forward	AGACGGGAATTTGATGTTTGAC
		Reverse	TGTCAATCAGGGCTCTCTCC
*Forkhead box protein O1* (*Foxo1*)	NM_001285835.1	Forward	ACCGTATCTGTGTGTGTGTGTG
		Reverse	ACAGCCAAGTCCATCAAGAC
*Glyceraldehyde-3-phosphate dehydrogenase* (*Gapdh*)	NM_007393.5	Forward	TGGACCTCATGGCCTACATG
		Reverse	AGGGAGATGCTCAGTGTTGG

Note: *Gapdh* was used as a reference.

## Data Availability

Not applicable.

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
