# Peer review of "The Hepatic Antioxidant System Damage Induced with the Cafeteria (CAF) Diet Is Largely Counteracted Using SCD Probiotics during Development of Male Wistar Rats"

_nutrients, 2023, doi:10.3390/nu15214557_

Round 1
Reviewer 1 Report
Comments and Suggestions for Authors
The article entitled "Damage by cafeteria (CAF) diet on hepatic antioxidant system and the damage preventive capacity of SCD Probiotics in juvenile wistar rats" the authors express the beneficial effects of probiotics are attributed to their metabolite content with antioxidant properties or the postbiotic metabolites produced by them. In this study the authors explore the effects of different dietary practices on the hepatic antioxidant system of rats in developmental period were investigated by real-time PCR, enzyme activity measurements, and quantitative measurements of metabolites. The results shows that SCD Probiotics supplementation could reverse hepatic antioxidant system at genes and protein levels. However, there are various questions and major flaws needs to be addressed:
1. The authors did not see any significant change in the body weights, how did they establish obesity in order to see the changes in all of the tested parameters?
2. The authors should have atleast done some fecal microbiome analysis to compare between the groups?
3. CAF needs to be elaborated first time in the subject
4. ROS abbreviation needs to be expanded and can be used else where in the article
5. What is SCD Probiotics? Expand
6. How much of the tissue was taken for each of the assays?
7. MDA needs to placed next to Malonaldehyde?
8. Table 2. Gene names other than the gene codes to be given in a separate column?
9. All figure to adapt the each data point for each specimen in the bar graph?
10. Figure legends to have "n " number of tissues used for each assay?
11. Apart from mRNA expression of genes it would be of much importance of the upstream protein expression of transcription factors such as Nrf2 as well as down stream signaling SOD and Catalase? Protein expression with western blot results will eventually show us the transcription from gene to protein!!
Comments on the Quality of English Language
N/A
Author Response
Response to Reviewers
We thank the reviewers for taking the time to read our manuscript and providing valuable comments. We revised the manuscript based on the reviewers’ evaluations and addressed each helpful comment as outlined point by point below. The relevant changes and information have been added to the revised manuscript as red-colored text. In addition, the manuscript has undergone a comprehensive revision, encompassing both essential modifications and thorough linguistic evaluations. We hope that our revised manuscript is satisfactory for publication.
Reviewer(s)' Comments to Author:
Reviewer: #1
Comments and Suggestions for Authors
The article entitled "Damage by cafeteria (CAF) diet on hepatic antioxidant system and the damage preventive capacity of SCD Probiotics in juvenile wistar rats" the authors express the beneficial effects of probiotics are attributed to their metabolite content with antioxidant properties or the postbiotic metabolites produced by them. In this study the authors explore the effects of different dietary practices on the hepatic antioxidant system of rats in developmental period were investigated by real-time PCR, enzyme activity measurements, and quantitative measurements of metabolites. The results show that SCD Probiotics supplementation could reverse hepatic antioxidant system at genes and protein levels. However, there are various questions and major flaws needs to be addressed:
Comment 1: The authors did not see any significant change in the body weights, how did they establish obesity in order to see the changes in all of the tested parameters?
Response: In our recent study, we embarked on a journey to understand the impacts of diet on young mice, specifically during their crucial developmental stage, from the 21st to the 56th day after birth. During this period, which starts right after weaning, we aimed to analyze the effects of a cafeteria diet combined with SCD Probiotics supplementation on their liver tissue's antioxidant systems.
It's noteworthy to mention that our study's primary focus was to simulate a more realistic environment. Instead of resorting to specially produced high-fat diets, which might not mimic real-world scenarios, we opted for the cafeteria diet. The idea behind this choice was to ensure that the young mice would naturally gain obesity within the stipulated period, making our findings more applicable to real-world situations.
While our study zeroes in on the development period between the 21st and 56th days, it's worth highlighting that we have an ongoing project using rats. In this complementary study, our goal is to induce obesity and diabetes using the same cafeteria diet approach. If the committee finds it beneficial, we are more than willing to share the results from this parallel study.
We genuinely hope our study and its methodology resonate with the importance of understanding diet's impact during developmental stages. We believe our approach, combined with our findings, will offer a significant contribution to this field of study.
Comment 2: The authors should have at least done some fecal microbiome analysis to compare between the groups?
Response: Upon the completion of our experimental phase, we meticulously collected and labeled cecum content and stool samples from the sacrificed control and experimental rat groups. The intention was to examine the intestinal microbiota profiles through whole genome sequencing or metagenomic analysis. These samples were preserved in a minus eighty-degree cabinet, awaiting the appropriate moment for analysis.
Regrettably, due to current financial constraints faced by our institution, particularly in light of the soaring inflation in our country, we were unable to undertake the metagenomic analyses as originally planned. As these sequence analyses are priced in dollars, the requisite funds—approximately $4,000—proved prohibitive for our study's budget. This has necessitated a pause, as we await a more financially viable opportunity to complete this aspect of our study.
Nevertheless, in a separate investigation involving 24-month-old male Sprague-Dawley rats, we paired SCD Probiotics with TUDCA. The results of this study were illuminating. We observed a significant augmentation in the alpha diversity indices of the intestinal microbiota after a 30-day administration of SCD Probiotics. Concurrently, we noted a marked reduction in the Firmicutes to Bacteroidetes (F/B) ratio, a known marker for dysbiosis and various metabolic disorders. We believe these findings are pivotal and have incorporated them into our discussion section for added depth and context. Page 14-15, Lines 453-480.
We trust that despite the unforeseen challenges in our primary study, the added insights from our separate investigation will enrich our contribution to the field.
Comment 3: CAF needs to be elaborated first time in the subject
Response: We've made the necessary correction in the manuscript. Page 1, Lines 10.
Comment 4: ROS abbreviation needs to be expanded and can be used else where in the article
Response: In line with your suggestion, the full name of ROS is written in the first place used in the introduction section. Page 1, Line 40.
Comment 5: What is SCD Probiotics? Expand
Response: In our studies, we have chosen SCD Probiotics as the designated probiotic product. It's essential to clarify that "SCD" is not an acronym; rather, the company has elected to represent its product line in this particular manner. Given its commercial nature, we consistently refer to it as "SCD Probiotics" throughout the manuscript to ensure clarity and accurate representation.
Our research employed the Liquid Probiotic Supplement (Essential Probiotics XI - 500 ml, H.S. Code: 2206.00.7000) from the SCD Probiotics company. The key microbial constituents of SCD Probiotics are:
Bacillus subtilis, Bifidobacterium bifidum, Bifidobacterium longum, Lactobacillus acidophilus, Lactobacillus bulgaricus, Lactobacillus casei, Lactobacillus fermentum, Lactobacillus plantarum, Lactococcus lactis, Saccharomyces cerevisiae Streptococcus thermophiles.
Comment 6: How much of the tissue was taken for each of the assays?
Response: Thank you for this critical notice. In line with your suggestion, necessary corrections and updates were made under the heading "Experimental design and animal care" in the material and method section. Page 3, Lines 106-108. Page 3, lines 120-121.Page 5, Lines152-153.
Comment 7: MDA needs to placed next to Malonaldehyde?
Response: In line with your critical notice, the relevant correction was made as “Malondialdehyde (MDA) in heading “"2.3 determination of lipid peroxidation" section. Page 3, Line 119. Additionally, the abbreviation of malondialdehyde was used as “MDA” in later uses.
Comment 8: Table 2. Gene names other than the gene codes to be given in a separate column?
Response: In the line of your suggestions, the full names of genes were written without creating a new column to preserve Table 2 and manuscript template compatibility. Page 4.
Comment 9: All figure to adapt the each data point for each specimen in the bar graph?
Response: We reflected the relevant correction to all figures
Comment 10: Figure legends to have "n " number of tissues used for each assay?
Response: In the line of your suggestions, number of tissues used for each assay were added to each figure legend.
Comment 11: Apart from mRNA expression of genes it would be of much importance of the upstream protein expression of transcription factors such as Nrf2 as well as down stream signaling SOD and Catalase? Protein expression with western blot results will eventually show us the transcription from gene to protein!!
Response:
Thank you for your insightful comments and valuable suggestions regarding our study.
We wholeheartedly concur with your perspective on the significance of assessing upstream protein expression of transcription factors such as Nrf2, as well as downstream signaling components like SOD and Catalase. Indeed, assessing protein expression through techniques like western blotting would provide a comprehensive view from gene transcription to protein translation, enriching our understanding of the mechanisms at play.
However, we would like to humbly mention that this study was carried out with our own limited resources. Unfortunately, due to budgetary constraints, we were not in a position to acquire the necessary consumables and equipment required for western blot analysis. While this was a limitation of our current research, we acknowledge the potential depth and clarity such an analysis would bring to our work.
We sincerely hope that you understand our position, and we are optimistic about the possibility of incorporating these methods in future research, should resources allow.
Thank you once again for your understanding and for highlighting this important aspect of our research.
Reviewer 2 Report
Comments and Suggestions for Authors The main topic topic of the manuscripi is the damage of Cafeteria on hepatic antioxidant system. The topic is original and relevant in the field. It addresses a specific gap in the field. The paper adds original results compared with other published material. The Authors should add a new chapter considering the study of serum sp-NOX2 and urinary 8-iso-PGF" alpha to better clarify the role of oxidant stress. The Conclusions should better discuss the deep study of oxidant stress. The references are appropriate. The Tables and Figures are clear and complete.Comments on the Quality of English Language
Moderate editing of English language is required
Author Response
Reviewer: #2
Comments and Suggestions for Authors
The main topic of the manuscript is the damage of Cafeteria on hepatic antioxidant system. The topic is original and relevant in the field. It addresses a specific gap in the field. The paper adds original results compared with other published material. The Authors should add a new chapter considering the study of serum sp-NOX2 and urinary 8-iso-PGF" alpha to better clarify the role of oxidant stress. The Conclusions should better discuss the deep study of oxidant stress. The references are appropriate. The Tables and Figures are clear and complete.
Comment 1: The Authors should add a new chapter considering the study of serum sp-NOX2 and urinary 8-iso-PGF" alpha to better clarify the role of oxidant stress.
Response:
Thank you for your thoughtful feedback.
I'd like to clarify a point: while we did collect stool samples to analyze intestinal microbiota after the animals were sacrificed, we unfortunately did not collect urine samples in this study.
Furthermore, we've conducted analysis on liver enzymes from the serum samples, but those findings are part of another study which is currently under peer review. We thought it prudent not to include those results in this manuscript to avoid overlap or potential discrepancies. However, if it would be helpful, we're more than willing to share those preliminary results with the referees for better context.
We appreciate your understanding and patience in this matter.
Comment 2: The Conclusions should better discuss the deep study of oxidant stress.
Response: In line with your suggestion, in the conclusion section, oxidative stress was discussed in more depth at the cellular level by establishing a relationship with the bacterial microbiota, and the purpose of the study was emphasized again and explained more clearly by touching on the effects of the components. The relevant discussion is highlighted in red on page 12 and page 14.

Round 2
Reviewer 2 Report
Comments and Suggestions for Authors
The Authors answered correctly to all my queries
Comments on the Quality of English LanguageModerate editing of English language is required